# Social Factors Associated with the Effectiveness of a Spanish Parent Training Program—An Opportunity to Reduce Health Inequality Gap in Families

**DOI:** 10.3390/ijerph17072412

**Published:** 2020-04-02

**Authors:** Noelia Vázquez, Pilar Ramos, M.Cruz Molina, Lucia Artazcoz

**Affiliations:** 1Public Health Agency of Barcelona, Plaça Lesseps, 1, Barcelona, 08023 Barcelona, Spain; pramos@aspb.cat (P.R.); lartazco@aspb.cat (L.A.); 2Institute of Biomedical Research (IIB-Sant Pau), 08041 Barcelona, Spain; 3Department of Methods of Research and Diagnosis in Education, University of Barcelona, 08035 Barcelona, Spain; cmolina@ub.edu; 4CIBER Epidemiology and Public Health (CIBERESP), 28029 Madrid, Spain; 5Center for Research in Occupational Health, University of Pompeu Fabra, 08002 Barcelona, Spain

**Keywords:** children, parenting, psychological intervention, health, social inequalities

## Abstract

Parent training programs (PTPs) have been used extensively in Anglo-Saxon countries, but less so in Southern Europe. Several characteristics of families have been linked to effective parenting and positive development of children, but few studies have examined the social determinants of the effectiveness of PTPs. The Parenting Skills Program for families (PSP) is a PTP from Spain. This study aimed to identify the social characteristics (sex, age, country of birth, marital status, educational level, and employment status) of parents that determine the success of the PSP in relation to social support, parenting skills, parental stress, and negative behaviors among children. A quasi-experimental study with a prepost design with no control group was used. We conducted a survey before (T0) and after the intervention (T1). Sample size was 216. We fit multiple logistic regression models. Parenting skills increased more among parents with a lower educational level. Parents’ stress decreased more among parents who had a lower educational level, were unemployed, and were men. Social support increased among parents who were younger, unemployed, or non-cohabiting. We found no significant differences in the effect on children’s negative behaviors according to the social factors evaluated. The PSP is effective for socioeconomically diverse families, but the success differs according to the parents’ social profile. Unlike most previous studies, the results were better among more socially disadvantaged people, highlighting the potential of this kind of intervention for reducing the social inequality gap between groups.

## 1. Introduction

Abundant evidence underscores the value of parenting practices that promote children’s well-being [1]. Parent training programs (PTPs) aim to improve the positive development of children and adolescents by giving parents or primary caregivers knowledge about child development, and teaching them skills [2] for effective parenting (e.g., praise, appropriate discipline, monitoring) [3]. This type of intervention has been extensively used and evaluated in Anglo-Saxon countries (Canada, Australia, USA, and UK), with positive effects on several variables related to learning and social and health outcomes (for example, stress) among children and parents, and with inconsistent effects for other variables (for example, self-esteem) [4,5]. While these interventions have become increasingly popular in Southern European countries in recent years, most studies have serious limitations, such as lack of rigorous program design based on explicit conceptual frameworks, standard implementation, and evaluation [6].

In the context of the “Health in the Neighbourhoods” community health strategy [7] in Barcelona (Spain), and partially related to the economic crisis that started in 2007, increasing problems between parents and children were detected, including family conflicts, teenage pregnancies, and high alcohol consumption among youths. After reviewing different international, national, and local datasets and websites (Pubmed, Centers for Disease Control and Prevention (CDC), Spanish Health Ministry, Catalan Health Service, etc.) and confirming that no systematic parenting skills interventions were being implemented in the city, our agency developed a new intervention, the Parenting Skills Program for families (PSP) [8,9], based on a previous one named Program-Guide to Develop Emotional Competences that had been implemented in some regions of Spain [10,11]. We added to the original PTP some new components that were recommended by CDC, like sessions with children participation and time-out strategy for children [12]. This group-based parent training intervention PSP was developed, implemented, and evaluated according to standards of effectiveness [12] using quantitative and qualitative methods and showed positive effects on parenting skills, parental stress, social support, and children’s behaviors [13,14,15].

Although differences in effective parenting have been linked to parents’ age, gender, and contextual and individual socioeconomic factors, such as educational level, family status, or employment status [16,17,18], few studies have investigated sociodemographic factors associated with the effectiveness of PTPs. The effectiveness of PTPs varies according to participants’ sociodemographic characteristics, as we can see in previous studies that have reported that the most vulnerable families benefit the least and obtain the poorest results after taking part in similar interventions [19]. For example, family adversity can undermine the efficacy of PTPs by disrupting training processes and the implementation of recommendations. Parents with low socioeconomic status (SES), those who are lonely or young, have unstable housing, or rely on government subsidies generally have poorer outcomes. Nevertheless, an increasing number of studies in recent years have described that the most vulnerable families benefit the same or more from such interventions [20]. It is necessary to understand the reasons for such inconsistencies. In spite of the growing recommendations to make available universal programs with a public health perspective that explicitly take into account the role of sociodemographic factors [21,22], most evidence comes from the evaluation of programs oriented to vulnerable groups. In addition, inconsistencies among studies can derive from difficulties in comparing predictable variables due to their differing categorization [23]. These inconsistencies also can be explained because most of those examining individual characteristics that may influence the effectiveness of PTPs generally examined only a few factors of parents’ unfavorable situations. Moreover, most of these studies have been carried out in Anglo-Saxon countries [2,19], few of them with a universal perspective [24] and, to our knowledge, there are no studies carried out in Mediterranean countries, which are very different in terms of family models and cultural context [25].

This study presents an opportunity to increase the knowledge about the link between parents’ social profile and the effectiveness of PTPs by examining an intervention with universal perspective carried out in a Spanish city. This PTP was carried out ensuring participant heterogeneity with respect to levels of vulnerability and the study selected as predictor variables social factors related to social determinants of health framework [26]. Its objective was to identify the parents’ social characteristics associated with PSP success for four outcomes: social support, parenting skills, parental stress, and negative behaviors among children.

## 2. Materials and Methods

### 2.1. Research Design and Sample

This is a quasi-experimental study with prepost design (T0 and T1), with no control group. At the beginning of the intervention, the sample comprised 257 parents (87.2% women), of whom 216 completed the questionnaire both at baseline and immediately after the intervention (84% retention rate).

### 2.2. Ethical Considerations

Participants received a document explaining the research process and gave their authorization by signing a consent form. To ensure confidentiality and according to data protection laws and regulations, we created two datasets, one containing identifying information and a code assigned to each participant, and another with no identifying information and with the code and the data for the different variables of interest. The research procedure was consistent with the guidelines for good research practices from the Barcelona Public Health Agency and the University of Barcelona and met all relevant legal requirements.

### 2.3. Description of the Intervention

The program consisted of 11 sessions with 12–14 parents, grouped according to their children’s age (2–6 years old, 7–11 years old, and 12–17 years old). If parents had more than one children with different ages, at home they could apply the learned parenting strategies to all their children, but it was more useful for the children of the group selected age. The main contents were different dimensions of positive parenting (for example, parent development stages understanding, parent and children self-esteem, assertiveness, or active listening and empathy). The core components of the program were the same for all parents, but there were some material differences depending on the children’s age (for example, role-playing or cases were adapted to children’s age). The sessions were delivered by professionals from social, health, or educational services previously trained by professionals from the Public Health Agency of Barcelona.

### 2.4. Data Collection

We used a self-administered questionnaire to collect data on the four outcome variables, and we also collected sociodemographic data on sex, age, country of birth, marital status, educational level, and employment status. The PSP was piloted between 2011 and 2013 [11] and evaluated between 2013 and 2015, with the participation in PSP sessions of 257 parents grouped into 24 groups according to children’s ages [13,14,15].

### 2.5. Dependent Variables

We examined four outcomes: social support, parenting skills, parental stress, and children’s negative behaviors. Social support was measured using the Duke-UNC Functional Social Support Questionnaire (DUFSS) [27] consisting of eight items divided into two subscales (confident [α = 0.92] and emotional [α = 0.81] social support) and scored on a five-point Likert temporal scale. The score range was between 40 (high social support) and 8 (low social support). Parental skills were measured using 19 items distributed in six dimensions and scored on a four-point Likert temporal scale, which was previously translated into Spanish and validated [28]. Its subscales were: (1) children’s understanding, (2) emotional self-regulation, (3) parenting self-esteem, (4) empathetic and assertive communication, (5) agreements, and (6) behavioral regulation. The Cronbach alpha was between 0.69 for subscale 6 and 0.89 for subscale 4. The score range was between 57 (high parenting skills) and 0 (low parenting skills). Parental stress was measured through the Spanish version of the Parental Stress Scale [29] consisting of 12 items distributed in two subscales (Baby’s Rewards [α = 0.77] and Stressors [α = 0.76]) scored on a five-point Likert scale (degrees of agreement/disagreement) (points scored between 1 and 5). The score range was between 60 (high parental stress) and 0 (low parental stress). Data on behavioral problems in the children were collected using the five-item subscale of the Strength and Difficulties Questionnaire [30], where items are scored on a three-point Likert scale (α = 0.83). The score range was between 15 (high negative children behaviors) and 0 (low negative children behaviors). Following similar studies [31,32,33], these four outcome variables were dichotomized by calculating the difference between the pre- and postintervention scores for each variable, and using the median as the cut-point. Parenting skills and social support were considered successful when the difference was above the median, and parental stress and children’s negative behavior when the difference was below the median.

### 2.6. Predictor Variables

The predictor variables were: (1) gender (woman vs. man), (2) age (four categories, <30, 30–35, 36–40, and >40 years old), (3) country of birth (born in Spain vs. born in another country), (4) marital status (married or cohabiting vs. non-cohabiting), (5) educational level (up to primary/elementary school studies completed, secondary/high school studies completed, or university studies completed), and (6) employment status (unemployed, housewife, employed, or other).

### 2.7. Statistical Analysis

Regarding the scales, pre- and immediate postintervention item scores for each outcome variable were summed and medians calculated. The Kolgomorov–Smirnov test confirmed a non-normal distribution of data. We tested for bivariate association between the predictors and outcome variables using the Pearson chi-square test. We included medians and 95% Confidence Interval (CI) for each variable. Finally, we fit multiple logistic regression models. We included adjusted Odd Ratio (aOR) and 95% Confidence Interval (CI) for each variable. We used the Wald test to test for a linear trend between outcomes and age and educational level. Missing data from sociodemographic variables were excluded from the bivariate and multivariate analysis. All analyses were carried out using Statistical Package for the Social Sciences (SPSS) version number 20.

## 3. Results

### 3.1. Description of the Sample and Attrition

Sociodemographic characteristics of the study population are described in Table 1. Most participants had one child (between 34% and 38%) or two children (between 43% and 47%), most of them were aged 3–5 years, and 50% were girls. Sixteen percent did not complete the intervention mainly due to work-related factors (e.g., change in work schedules or finding a job), family reasons (having a new child or caring for a dependent relative), and moving to a new location. We observed statistically significant differences in level of education between pre- and postintervention, with people with secondary studies being more likely to drop out.

### 3.2. Relationship between Parents’ Sociodemographic Characteristics and Effectiveness of the Parenting Skills Program for Families (PSP)

Results from bivariate analysis, described in Table 2, identify outcome improvement according to parents’ social factors. In most cases, the more disadvantaged groups (parents who were men, younger, immigrants, non-cohabiting, unemployed, or possessed a low level of education) had poorer preintervention medians and more favorable postintervention results. Men were more likely to reduce parental stress levels (80.0%), and younger parents more likely to improve parenting skills (66.7%), perceive social support (69.0%), and reduce more parental stress (69.0%). Non-cohabiting parents (73.9%) were more likely to improve social support. Parents with lower educational level showed a greater improvement in parenting skills (66.7%), a reduction of parental stress levels (70.2%), and negative behaviors in their children (70.4%). Finally, unemployed parents were more likely to have improved social support (67.7%) and reduced parental stress (64.6%) (Table 2).

Results from multivariate analysis, described in Table 3, allowed us to identify association between parents’ social factors (sociodemographic characteristics) and outcomes related to the PSP (parenting skills, parental stress, social support, and children’s negative behavior). Educational level was the only predictor of improvement of parenting skills after participating in the PSP. Parents with primary education or less (aOR = 5.11, 95%CI = 1.90–13.60) and those with secondary studies (aOR = 3.45, 95%CI = 1.55–7.80) were more likely to have improved parenting skills than those with a university education, with a significant linear trend (Wald test). Parental stress was more likely to decrease among men (Ora = 3.62, 95%CI = 1.17–11.18), parents with lower educational level (aOR = 3.16, 95%CI = 1.19–8.56), and unemployed parents (aOR = 1.00) compared to housewives (aOR = 0.42, 95%CI = 0.23–0.92). We also observed a significant linear trend for educational level. We did not find any predictor variables that were significantly associated with improved negative behaviors among children, although there was a significant linear trend with age. Finally, perceived social support improved significantly in younger parents compared to parents aged 36–40 (aOR = 0.27, 95%CI = 0.09–0.80) and >40 (aOR = 0.28, 95%CI = 0.09–0.85), with a significant linear trend. Similarly, non-cohabiting parents (aOR = 3.46, 95%CI = 1.58–7.58) were more likely to have better perceived social support than married or cohabiting parents. Unemployed parents were more likely to have improved social support than workers (ORa = 0.40, 95%CI = 0.9–0.88) and pensioners or disabled individuals (aOR = 0.24, 95%CI = 0.06–0.89) (See Table 3).

## 4. Discussion

As far as we are aware, this is the first study carried out in Spain that reports an association between parent social factors and the effectiveness of a universal PTP on four outcomes. Moreover, we observed an equal or, in some cases, even greater improvement among more vulnerable parents with a more unfavorable starting point than among privileged parents. Therefore, we report four main findings: (1) parents with lower educational level showed the greatest improvement in parenting skills; (2) parental stress levels decreased most among men, unemployed parents, and those with a lower educational level; (3) social support improved most among unemployed parents, single parents, and young parents; and (4) we found no significant associations with parents’ social profile and negative behaviors in children.

Most previous studies about PTPs, similar in content and components to our program, carried out in other countries [19,34,35] found that socioeconomically disadvantaged families (unemployed or with a lower educational level) show poorer results after participating in PTPs. This finding was mainly related to the higher drop-out rate among parents with lower educational level [36]. However, a growing number of studies have found equivalent [37,38,39] or even better results [39] among the most socioeconomically disadvantaged families compared to the most privileged families. So, our findings and these previous studies highlight that PTPs could potentially reduce the gap between advantaged and disadvantaged families. This could be associated with cultural differences or the participants’ heterogeneity. Most evidence on the effects of PTPs comes from interventions similar in content and components, but not in the participants’ socioeconomic profile. Most previous interventions were addressed mainly to disadvantaged participants, such that samples show a very homogeneous socioeconomic status (i.e., regarding educational level or employment status), making it difficult to find statistically significant differences [40]. In addition, they were promoted and delivered only by non-universal services. In this regard, we recommend that this type of program be promoted as a free public service with emphasis on recruiting and consolidating the participation of families of heterogeneous socioeconomic profiles [41]. Previous studies have reported that high-risk families benefit more from diverse groups, probably because they see that ‘another world is possible’ [42], and because these parents actively seek help in PTP groups [43].

In addition to employment status or educational level, other variables such as gender, loneliness, and age have scarcely been explored by previous studies. Most PTP studies find no significant gender differences probably because the sample was mainly composed of women [44]. Some studies have identified lonely parenthood as a predictor of improvement in formal social support and in solving personal problems [45]. These results are consistent with most of our findings on improved social support among non-cohabiting parents, which also include single-parent families. Probably this is because non-cohabiting parents tend to feel more isolated [46], including in parenting. Studies reporting that the effectiveness of PTPs differs according to age have focused on the first months of baby life [47]. They show that younger parents, many of whom are new parents in the transition to parenthood, need the support [48] that these sessions provide. This evidence is consistent with our findings. Peer support, facilitator support, and the program contents are more novel for younger new parents than for experienced parents who may have participated in similar sessions or programs previously.

### 4.1. Limitations

One limitation of this study is the lack of data on the relationship between the intervention dosage and its effects. We observed statistically significant differences in educational level between participants pre- and postintervention, with people with secondary studies being more likely to drop out, but we do not know the reasons. We know that the main reasons for leaving the program are family or professional issues. Maybe there could be an association between participants’ educational level and these issues. In addition, we were unable to determine whether participants who attended fewer sessions had the same, better, or poorer results than those who attended more sessions. We were also unable to evaluate the possible effects on the outcomes of some aspects related to the implementation of the program, such as professional training.

Another limitation is that the variable “children’s negative behavior” could be affected by ceiling effects [49]. The range of improvement between T0 and T1 for all groups is narrow, so it is difficult to find differences between groups.

Finally, another limitation is that we did not use a randomized control trial (RCT) design with a control group. This type of intervention is still rare in Spain, and a control group was not feasible. Thus, we used a quasi-experimental design with no control group, as this is the recommended approach when proper randomization is not possible. Nonetheless, it has been suggested that an RCT is not required if an intervention is well developed, meets generally accepted standards, and demonstrates changes in evaluations with no control groups, and when qualitative methods highlight the mechanisms involved [50]. In this sense, it will be important to collect qualitative data in future studies to improve our understanding of how family social determinants and program implementation influence the effects of PTPs.

### 4.2. Practical/Research Implications of the Findings and Recommendations

These findings highlight several implications and recommendations:Although PTPs are not as common in Southern European countries as in Anglo-Saxon countries, this study shows its effectiveness for different outcomes in a Spanish city.This type of program suggests the need to promote PTPs by public bodies in a universal manner, ensuring recruitment of heterogeneous types of families and not only the most vulnerable families.It seems also important to promote them as a no-charge territorial public service.PTPs should suit all parents equally regardless of their sociodemographic background.More randomized control trials with larger sample sizes are needed. Moreover, studies comparing the effectiveness in different countries are needed in order to understand the role of policy and cultural factors.

The role of the implementation process for the success of the interventions deserves further attention.

## 5. Conclusions

This study has shown that a PTP based on the current scientific evidence is effective in a Southern European city. Disadvantaged and privileged parents improved different parenting baseline outcomes after participating in PTP sessions. Contrary to most previous research, we found that more disadvantaged families were more likely to improve, possibly due to cultural differences and the heterogeneity of the participants. The gap between disadvantaged and privileged families was reduced. Our findings are consistent with recommendations to extend these interventions to the general population, and not only to high-risk families. Our study highlights the potential of PTPs as an intervention for reducing the social inequality gap, maintaining the likelihood of more socioeconomic privileged parents benefiting from PTPs described in previous studies, but also improving the likelihood of more vulnerable parents benefiting also from it. Future studies should tackle the challenge to improve PTPs to suit all parents equally regardless of their sociodemographic background and close this gap.

## Figures and Tables

**Table 1 ijerph-17-02412-t001:** Sociodemographic characteristics of the study population before and after intervention.

	T0		T1		Drop-out
	N	%	N	%	%
Total	257	100	216	100	16
Sex					
Women	224	87.2	191	88.4	14.7
Men	33	12.8	25	11.6	24.2
Age					
<30	50	19.5	42	19.4	16
30–35	64	24.9	55	25.5	14.1
36–40	81	31.5	65	30.1	19.8
>40	60	23.4	52	24.1	13.3
Missing	2	0.8	2	0.9	
Country of birth					
Spain	152	59.1	131	60.7	13.8
Other countries	104	40.5	84	38.9	19.2
Latin America	51	19.8	40	23.6	21.6
Africa	22	8.6	20	10.2	9.1
Europe	16	6.2	14	7.4	12.5
Others	15	5.8	10	6.9	33.3
Missing	1	0.4	1	0.5	
Marital status					
Married or cohabiting	196	76.3	165	76.4	15.8
Other	59	23	49	22.7	16.9
Missing	2	0.8	2	0.9	
Educational level					
University	92	35.8	79	36.6	14.1
Secondary/high school	90	35	68	31.5	24.4 *
Primary/elementary school or lower	61	23.7	57	26.4	6.6
Missing	14	5,5	12	5.6	
Employment status					
Unemployed	80	31.1	65	30.1	18.8
Employed	105	40.9	89	41.2	15.2
Housewives	50	19.5	45	20.8	10
Others	20	7.8	15	6.9	25
Missing	2	0.8	2	0.9	

T0: baseline; T1: immediately after the intervention/* *p* < 0.05 (Chi-Square test).

**Table 2 ijerph-17-02412-t002:** Outcome improvement according to parents’ social factors. Bivariate analysis.

	Parenting skills (57) ****					Parental Stress (60) ****					Children’s Negative Behaviors (15) ****					Social Support (40) ****				
	T0	CI	T1	CI	N (%)	T0	CI	T1	CI	N (%)	T0	CI	T1	CI	N (%)	T0	CI	T1	CI	N (%)
Sex																				
Women	36	35–38	40	28–39	89 (46.6)	27	24–32	23	22–28	90 (49.2)	3	1–4	2	1–4	90 (50.8)	32	19–35.5	35	16–39	90 (4.7)
Men	32	0–34	50	32.2–57	16 (64.4)	31	25.5–60	15	12–29	17 (80.0) **	4	3–10	0	1.5–2.5	17 (68.0)	24	8–30	38	28–40	16 (64.0)
Age																				
<30	32	3–38	47	31–55	28 (66.7) *	35	24–59	20	14–28	29 (69.0) *	5	1–9	1.5	0–4	25 (65.8)	23	9–26	38	23–40	29 (69.0) *
30–35	37	32.5–39	41	29.5–42	24 (43.6)	28	23–38	24	22.5–33.5	29 (52.7)	3	1.5–5.5	2	1.5–5	30 (56.6)	33	19–37	37	30.5–39.5	29 (52.7)
36–40	36	25–40.5	44	34.1–55.5	34 (52.3)	25	21–38.9	22	13.5–28	35 (53.8)	2	1–4	2	0–3	31 (51.7)	33	18–38	36	30–37	25 (38.5)
>40	37	28–44	39	27–40.5	18 (34.6)	26.5	21–31	23	21–37	19 (36.5)	2	1–4	2	1–3	19 (38.8)	30.5	18.6–36.5	34	17–36.5	22 (49.1)
Country of birth																				
Spain	36	33.5–42	40	31–55.5	58 (44.3)	27	21.5–31	23	13.5–21	71 (54.2)	2	1.5–4	2	0.5–3.5	67 (54.5)	33	16.5–36.5	36	21–38.5	63 (48.1)
Other countries	35	20–37	43	32–42.5	46 (54.8)	29	29–41.5	22.5	21–37	43 (51.2)	3	2–7	2	0.5–5	39 (50.0)	30	13–34	36	27–40	42 (50.0)
Marital status																				
Married or cohabiting	36	31–44	40	28–57	77 (46.7)	28	17–33	23	13–32	87 (52.7)	3	1–4	2	0–3	88 (55.3)	32.5	14–37	36	16–38	69 (41.8)
Non–cohabiting	34.5	24–38	43	37.2–48	26 (53.1)	28	25–43	22	14–43	27 (53.1)	3	2–8	2	0–4	18 (42.9)	29	13–38	36	21–38	36 (73.5) ***
Educational level																				
University	39	30–40	40	32–42	23 (29.1)	26	21–39	24	22–37	33 (41.7)	2	3–7	5	2–5	37 (48.7)	34	13–39	36	27–40	34 (43.0)
Secondary	34	3–38	40	31–55	39 (57.4)	28	24–59	22	14–22	36 (52.9)	2	1–9	2	0–3	29 (46.0)	29	9–26	34	23–39	35 (51.5)
Primary or lower	29	5–45	45	36–54	38 (66.7) ***	36	27–59	18	12–24	40 (70.2) **	2	2–8	2	0–6	38 (70.4) *	21	10–40	38	23–40	31 (54.4)
Employment status																				
Unemployed	32	29–34	41	39–44	38 (58.5)	33.97	29–37	23	21–24	42 (64.6) ***	3	3–5	2	1–2	38 (64.4)	26	19–32	38	34–38	44 (67.7) **
Employed	37	36–39	41	40–43	35 (39,3)	26	25–27.5	24	22–25	37 (41.6)	2	2–2	2	1–2	39 (45.3)	33	30.5–35	35	33.5–37	34 (38.2)
Housewives	33.3	30–37	39.4	37–43	24 (53.3)	28.5	27–36	23	22–28	21 (46.7)	3	2–4	2	2–2	21 (51.2)	28.5	23–33	35	33–36	21 (46.7)
Others	37.5	18–40	39	33.5–54	7 (46.7)	27	26–46	22	14–23	13 (84.7) ***	3	2–7.5	1	0–3	8 (57.1)	33.5	12–36.5	36	26–38.5	6 (40)

Pearson Chi-Square test * *p* < 0.05, ** *p* < 0.01, *** *p* < 0.001; **** Maximum total score for each variable; T0: baseline medians; T1: immediately after intervention medians; 95% CI: confidence interval; N: number of people with improvement after the intervention; %: percentage of people from each group with improvement after intervention.

**Table 3 ijerph-17-02412-t003:** Multivariate analysis of association between parents’ social factors and outcomes related to Parenting Skills Program for families (PSP). Adjusted Odds Ratios (aOR) and 95% Confidence Intervals (CI).

	Parenting Skills		Parental Stress		Children’s Negative Behaviors		Social Support	
	aOR	95%CI	aOR	95%CI	aOR	95%CI	aOR	95%CI
Sex								
Women		1.00		1.00		1.00	1.00	
Men	2.73	1.00−7.51	3.62 *	1.17−11.18	2	0.72−5.50	2.5	0.93−6.69
Age								
<30	1.00		1.00		1.00a		1.00a	
30−35	0.64	0.23−1.81	1.37	0.46−4.07	0.88	0.29−2.66	0.54	0.18−1.59
36−40	1.43	0.51−4.00	1.39	0.49−3.93	0.81	0.28−2.38	0.27 *	0.09−0.80
>40	0.48	0.17−1.36	0.52	0.18−1.52	0.4	0.13−1.21	0.28 *	0.09−0.85
Country of birth								
Spain	1.00		1.00		1.00		1.00	
Other countries	1.21	0.63−2.30	0.78	0.41−1.50	0.71	0.37−1.36	0.84	0.43−1.61
Marital status								
Married or cohabiting	1.00		1.00		1.00		1.00	
Non−cohabiting	1.11	0.53−2.36	1.35	0.65−2.84	0.52	0.24−1.14	3.46 **	1.58−7.58
Educational level								
University	1.00c		1.00		1.00		1.00	
Secondary	3.45 **	1.55−7.80	1.57	0.71−3.43	0.89	0.41−1.94	1.04	0.47−2.28
Primary or lower	5.11 **	1.90−13.60	3.16 *	1.19−8.56	2.09	0.77−5.64	0.75	0.29−1.95
Employment status								
Unemployed	1.00		1.00		1.00		1.00	
Employed	0.86	0.39−1.94	0.63	0.31−1.20	0.65	0.29−1.45	0.40 *	0.19−0.88
Housewives	0.78	0.32−1.89	0.42 *	0.23−0.92	0.6	0.24−1.51	0.48	0.20−1.17
Others	0.43	0.12−1.54	3.1	0.63−16.74	0.54	0.15−1.96	0.24 *	0.06−0.89

Multivariate analysis * *p* < 0.05, ** *p* < 0.01; Linear trend with Wald test a *p* < 0.05, b *p* < 0.01, c *p* < 0.001.

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
