# Peer review of "Social Factors Associated with the Effectiveness of a Spanish Parent Training Program—An Opportunity to Reduce Health Inequality Gap in Families"

_ijerph, 2020, doi:10.3390/ijerph17072412_

Round 1

Reviewer 1 Report

Dear editor,

Thanks a lot for the possibility to do a second review of this paper.

I find it interesting and mostly well written.

The talk about groups at page 3(line 90)  is confusing, is it groups of parents trained?

I would like to understand that they cvreated groups with parents of children at different ages a little bit earlier in the paper

Some places the language could be improved, e.g. the author writes about improved parental stress when I think they mean that the level of stress has decreased (confirmed in the tables).

From line 256 at p.13 they write: It is crucial to increase randomized control trials with larger sample sizes. If they mean that more RCT's are needed they should write that.

Reviewer 2 Report

The authors of the current study aimed to explore the sociodemographic determinants of effective parent training programs (PTP) in Spain measured by improvement in parents’ social support, parenting skills, parental stress and negative behaviours among children. Developing an effective PTP is of importance as this may help parents to improve parenting skills and parental stress, which in turn can reduce their child’s negative behaviors. Associations were found between specific sociodemographic characteristics and parental stress, parenting skills and social support in favour of disadvantaged parents (i.e. men, young, low education, etc.) although no associations were found with child negative behaviors. These results suggest that parents with poorer sociodemographic characteristics are more likely to improve after the intervention compared to more privileged parents. Based on these findings the authors recommend using a universal program which includes heterogenous groups in terms of sociodemographic background. The question addressed in this manuscript is important but the authors should have drawn a different conclusion from the results.  

Abstract

  1. Lines 14-16: “While several characteristics of families have been linked to effective parenting and positive development of children, few studies have evaluated the social determinants of the effectiveness of PTP.” This sentence is not clear. The word “while” means the sentence needs to include contradiction but both parts of the sentence say the same thing.
  2. Line2 26-27: “…the potential of this kind of interventions for reducing social inequalities in health.” If you recommend using a universal program delivered to diverse groups which include advantaged parent who have lower chance to benefit from the program the inequality remains, it is now just against privileged parents. Please clarify.

Introduction

  1. Line 49: replace “included” with “added”.
  2. Line 53-56: please change social factors to sociodemographic characteristics/factors throughout the text. The two parts of the sentence should contradict each other but they are saying the same thing.
  3. Line 55-56: “The…. Characteristics.” The meaning of this sentence is unclear because it repeats the previous sentence. Did you mean to say that it is important to investigate the relationship between sociodemographic characteristics and program effectiveness? You then need to rephrase this sentence and link it to the next sentences.
  4. Line 66: add is before oriented.
  5. Lines 63-66: please see comment no. 28 and change these sentences accordingly. These sentences should also be moved to the end of the paragraph.
  6. Line 66-69: “In addition, inconsistencies among studies….” This sentence needs to be rephrased or divided to two separate sentences.
  7. Line 73: replace “represents” with “presents”.
  8. Line 77: please clarify what “social determinants of health framework” means.

Materials and methods

  1. Sections 2.6 and 2.7 should be moved backwards after research design and sample.
  2. Statistical analysis: this section needs to be more detailed. Please explain why using the tests. In addition, the outcome measures are expressed as the difference between pre- and post-intervention scores. The pre-intervention score of some outcome measures is poorer in disadvantaged groups. In order to rule out the possibility that the improvement seen in these groups is derived from poorer baseline scores, differences between groups should be evaluated separately for pre- and post-intervention using medians/means depending whether the data is normally distributed. The results of normality test should also be shown.

Results

  1. Line 145: replace “the study population…” with sociodemographic characteristics of the study population...”
  2. Lines 147-148: “16%” at the beginning of the sentence should be replaced with Sixteen percent and “[16%]” at the end of the sentence should be removed.
  3. Section 3.2 title should be “relationship between parents’ sociodemographic characteristics and effectiveness of the PTP.
  4. Line 157: statistical analysis should be described briefly.
  5. Lines 157-163: the results of the analysis indicate which group is more likely to improve and not which group had greater improvement. Therefore, these sentences should be rephrased accordingly. In the first brackets the meaning of % should be explained.
  6. Line 166: start this paragraph by explaining the purpose of the multivariate analysis and add that the results are presented in Table 3.
  7. Line 171: add “compared to midwives after “unemployed parents” and replace the OR with the midwives’ OR and IC95%.
  8. Lines 172 and 174: add “linear” before “trend” and “(Wald test)” and add the corresponding footnote on the Table.

Table 1

  1. Title should be “Sociodemographic characteristics of study population before and after intervention”.
  2. In some categories the size of the sub-samples does not sum up to the total number.

Table 2

  1. The values in “T0” and “T1” should be described. Are these medians?
  2. “N (%)” should be “N improved (%)”.
  3. Legend: add details about the statistical analysis before the p-values.

Table 3

  1. Legend: add “Multivariate regression analysis” before the significant p-values with the asterisks.

Discussion

  1. Lines 208-209: despite the difference in the likelihood of different groups to benefit from the intervention, the authors recommend using universal programs to target parents with heterogenous sociodemographic profiles. While it is important to provide equal opportunities to disadvantaged parents, it is also important not to discriminate advantaged parents. Therefore, the authors should address the question why would advantaged parents participate in such program if they cannot benefit as much as disadvantaged parents. I would think the conclusion should be that the program should be improved to suit all parents and to be beneficial to all parents regardless their sociodemographic background.

Limitations

  1. Lines 236-238: the baseline scores are already pretty low to start with so there was no place for large differences. In addition, there could also be a floor effect (not a ceiling effect) as men receive a score of zero post-intervention.

Conclusions

  1. Lines 265-266: “Our findings are consistent with recommendations to extend these interventions to the general population, and not only to high-risk families.” The findings of the current study show the opposite, which is that the program is more beneficial to disadvantaged parents but the presence of advantaged parents may be necessary for disadvantaged parents to improve. This is not a sufficient justification to include advantage parents in a program which in its current setup will not benefit them to the same extent as in disadvantaged parents. Please clarify.
  2. Lines 266-269: “Our study highlights the … and children’s negative behaviors …” the study does not show the potential of PTP to improve children’s negative behaviors as no significant difference between groups was found. “… reducing social inequity” this is not one of the aims of the program. This whole sentence is not accurate as the results do not point at the potential of the program, this can be explored only with a RCT. The study is looking at the likelihood of groups with different sociodemographic background to benefit from the program.   

Reviewer 3 Report

Social factors associated with the effectiveness of a Spanish parent training program. An opportunity to reduce social health inequities in families

The title of the article does not reflect the topic and the conclusions of the article. It does not explain how to reduce in inequalities; rather shows different outcomes depending on socioeconomic factors

I have some important methodological concerns:

The presentation of outcomes looks very problematic; the authors did not refer to some literature that suggest this type of dichotomization (line 109-112):

 “The outcomes were dichotomized by calculating the difference between the pre- and post-intervention scores for each variable, and using the median as the cut-point. Parenting skills and social support was considered successful when the difference was above the median, and parental stress and children’s negative behavior when the difference was below the median”

I suggest using another approach, “Reliable Clinical Change” (Guhn M., Forer B., Zumbo B.D. (2014) Reliable Change Index. In: Michalos A.C. (eds) Encyclopedia of Quality of Life and Well-Being Research. Springer, Dordrecht). In that case you can use “improved” and “recovered” as positive outcomes and then compare “before” and “after” measures. The results as they are presented are very difficult to interpret.

Other remarks

Abstract

The aim of the study (line 17-18):

This study aimed to identify the social characteristics of parents that determine the success of PSP in relation to social support, parenting skills, parental stress and negative behaviours among childre

The social characteristics associated with outcomes of PSP, not determined

Line 23: How the program can increase social support? The feeling of social support?

Introduction

There are many PTP available, even in Spain. The motivation to develop this particular PTP is weak.

 The aim of the study is unclear. How the social support could be an outcome of PTP?

Methods

As I sad previously, the determination of the outcomes is not sufficient. The outcomes must be clearly defined and all the analyses have to be re-run.

This is unclear how the missing data issues were addressed.

Results

Table 2.

All the results at T0 and T1 have to be presented with CI (confidence interval).

Discussion

Line3 193 - 194

….with parent’s social profile and negative behaviors in children.

Changes in negative behaviors in children?

Author Response

Please see the attachment. Thank you very much.

This manuscript is a resubmission of an earlier submission. The following is a list of the peer review reports and author responses from that submission.

Round 1

Reviewer 1 Report

Thank you for the opportunity to review the manuscript (Manuscript ID: ijerph-626913) titled, "Social factors associated with the effectiveness of a Spanish parent training program. An opportunity to reduce social health inequities in families.” This paper aimed to identify the socio-demographic characteristics of parents associated with improved outcomes on the PSP parenting program. Although the study benefits from a large sample size and is one of the first studies exploring participant characteristics associated with improved outcomes on a parenting program in Spain, I am not convinced about the results of the study primarily because of a small proportion of participants recruited from the minority groups and the possible ceiling effects contributing to non-significance of results for the other groups. The introduction needs further work in terms of strengthening of the study rationale, a careful proof read and checking of the manuscript for incorrect sentence structures and grammatical errors, as well as a section or discussion of ‘Research/ practical implications of the study findings’.

Below are some specific suggested areas for revision that could enhance the current manuscript.

Abstract

The abstract is generally well-written. In Line 20, please include a time-frame of post intervention. The sample size should be mentioned separately in another sentence. The manuscript uses T1 to describe the post-intervention time point, this needs to be mentioned in the abstract too.  

Introduction

The importance of this study is not clear from reading the intro, i.e. Why is this study needed if previous research has already looked at socio-demographic characteristics associated with positive outcomes for other parenting programs? How is PSP different from other programs and what implications does that have for future parenting research and practice?

Lines 34-35-The authors mention that parenting interventions have been used extensively in north European countries but then provide examples of non-European countries in brackets, e.g. Australia, Canada. The same sentence mentions ‘interesting effects’ on outcomes. Please clarify what does this refer to-positive outcomes/ inconsistent outcomes and why are these interesting? Line 43-44- The authors mention that it was confirmed that no systematic parenting interventions were in place in the city where the research was conducted. Please describe the process undertaken to confirm this-was a systematic review undertaken? What other sources were referred to in the process? Page 2, first paragraph-Please put all references in brackets. Line 45-It is mentioned that PSP was based on a previous program-Please provide a name of that program and details of that intervention and why was it decided to adapt PSP using that program. Are there any effectiveness data/ studies undertaken for that program? If yes, these must be cited too. Page 2, second paragraph- Please provide some explanation for why the socioeconomic factors investigated with previous parenting programs would be different to PSP? How is the PSP program different to other programs that have evaluated these factors? In the intro please provide more details for why these specific predictor variables and outcomes were chosen for evaluation in this study.  

Method

Description of the intervention: Please provide more details about the intervention/ PSP program. What is the content of the intervention? How do the sessions differ for different age groups? Research design, sample and data collection: Please include a separate Measures section to describe the outcome measures. Please also provide total scores, range of scores and direction of improvement for each measure, as this is crucial information for the interpretation of results provided in the paper. It is unclear if the alphas provided are from previous studies or the current sample. Please provide both. Ethical considerations-Lines 114-116. It is mentioned that two separate datasets were created. Please mention why this was done. Also although the authors mention that the study was conducted in accordance with the ethical guidelines, there is no mention of ethics approval for this study. Please mention this explicitly and the name of the HREC which provided approval for the study.

Results

Please include a separate section on attrition from the study. Lines 126-127 mention differential attrition based on education levels of the participants- Which tests were used to calculate this? Please present results clearly, i.e. what % dropped out from each group. Please also provide the effect sizes in the tables.

Discussion

Lines 165-166 and Table 2-The authors mention that more improvements were seen for the most vulnerable groups, especially in the context of their worse scores to start with. It is possible that significant improvements were not seen in the other groups because of the ceiling effects, i.e they were already doing relatively well and thus there was not much scope for further improvement. This is difficult to interpret without any reference to the total scores for outcome measures in the paper. Further, the results have shown that there was wide variation in scores at T0 where significant findings have emerged (e.g. for parenting skills, stress and social support) but not for child behaviours, which possibly contributed to significant results. The discussion will benefit from a section on the ‘Research/ Practical implications of the findings’ to make this paper more relevant for the readers of the IJERPH.

I hope the above feedback is useful in reviewing and revising the manuscript.

Reviewer 2 Report

This paper is important because of it's focus on how social factors may influence how families can benefit from participation in parent-training programs. It seems to be a follow-up of the paper with reference no. 13 in this paper, many similar descriptions are found in these to manuscripts (they should be compared because there is a risk of plagiarism). Even though the aim of the paper is interesting it must be improved before considered printed, my comments are given in relation to each main chapter.  

Introduction
The title refers to social factors that may affect the effectiveness of parent-training programs but the introduction does not answer to this - it talks mainly about what is done before in Spain and other places. Please, consider a more to the point introduction that may shed light on the current research-question. Some of the sentences are not understandable (e.g.  line 60-61). Materials and methods. 
You should report about design and participants/sample before presenting the intervention. Next, more information about the intervention is needed, it seem relevant as this paper focus on whom PTP may be most valuable for. Measures should be presented in a separate section and next, the presentation of statistics seem  insufficient. The use of dichotomized variables, using the median as a cut off point - is that appropriate? Results.
Descriptions of the sample is given twice, both in text and table, reconsider this. The levels of education is difficult to understand for persons outside Spain, could you use years of education instead of these three categories? Table 2 is interesting, but would be more readable if supported by more explanations. Discussion
The discussion of differences and similarities, between results in this study compared to others are difficult to understand. Does it only be about differences between study-samples, or may the design of this parent-training program affect disadvantaged families in new ways?

Reviewer 3 Report

More details are needed regarding the intervention curriculum. What are the key messages that are being promoted? How many parents are in these sessions? Who delivers these sessions? Who trains the interventionists? What is the theory of change?

In the introduction, the authors state that prior literature has been limited with not enough rigorous studies. However, the current evaluation approach of a quasi-experimental design, without a control group, is quite a weak study design. 

What is the rationale for examining these 4 particular outcomes?

It is unclear whether the study was powered from the outset to adequately determine statistical significance in the outcomes. 

Specifically, for the SDQ, more details are needed regarding the 5-item subscale. Which subscale was used and what is the rationale for this subscale as opposed to the other subscales or the full measure. 

There is no description of the missing data/loss to follow up - reasons for loss to follow up. This has implications on the findings, which are not appropriately considered in the analysis/discussion.

The Discussion is written and interpret results using a lot of causal language. Because this is a pre/post study design, it is critical that the interpretation is not causal. 

A greater discussion is needed about the components, content, and delivery strategy used in this intervention and its potential, or the implications of this specific intervention that is being evaluated.